# Mixture of Soft Prompts for Controllable Data Generation

**Derek Chen**♠  **Celine Lee**♡  **Yunan Lu**♠  **Domenic Rosati**†  **Zhou Yu**♠

♠ Columbia University,  † Scite AI,  ♡ Cornell University

{dc3761, yl4021, zy2461}@columbia.edu,
dom@scite.ai, cl923@cornell.edu

## Abstract

Large language models (LLMs) effectively generate fluent text when the target output follows natural language patterns. However, structured prediction tasks confine the output format to a limited ontology, causing even very large models to struggle since they were never trained with such restrictions in mind. The difficulty of using LLMs for direct prediction is exacerbated in few-shot learning scenarios, which commonly arise due to domain shift and resource limitations. We flip the problem on its head by leveraging the LLM as a tool for data augmentation rather than a model for direct prediction. Our proposed **M**ixture of **S**oft **P**rompts (MSP) serves as a parameter-efficient procedure for generating multi-attribute data in a controlled manner. Denoising mechanisms are further applied to improve the quality of synthesized data. Automatic metrics show our method is capable of producing diverse and natural text, while preserving label semantics. Moreover, MSP achieves state-of-the-art results on three benchmarks when compared against strong baselines. Our method offers an alternate data-centric approach for applying LLMs to complex prediction tasks.

## 1 Introduction

Complex natural language understanding (NLU) systems, such as semantic parsers, typically become useful only after training on copious amounts of labeled data (Chen et al., 2020). Due to the high cost of annotation, obtaining a sufficient supply of data quickly becomes infeasible. Low resource settings are particularly common when expanding a system into a new domain or service (Wang et al., 2015). This task of learning a target domain from limited data is referred to as domain-adaptive few-shot learning (Zhao et al., 2020).

Modern large language models (LLMs) have emerged as effective classifiers in low resource settings (Sanh et al., 2022; Chung et al., 2022), and

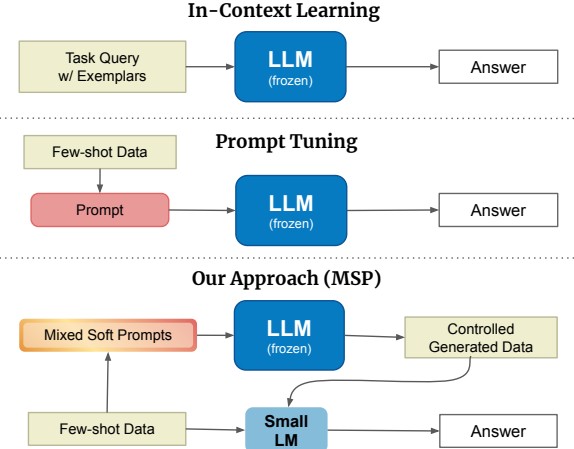

Figure 1: Standard in-context learning (top) directly prompts a frozen LLM with a query and exemplars. Prompt tuning (middle) trains a prompt that is then used to query the LLM. MSP (bottom) learns a set of soft prompts, mixes them together to generate attribute-preserving examples, then merges the augmented and original data to train a smaller, downstream model.

can even take advantage of few-shot examples without the need for gradient updates through in-context learning (ICL) (Brown et al., 2020; Xie et al., 2022). However, off-the-shelf LLMs have shown evidence of struggling with direct prediction in more complicated NLU tasks, such as those involving hierarchy or compositionality (Furrer et al., 2020; Qiu et al., 2022; Dziri et al., 2023). LLMs with ICL also exhibit problems when the target output requires a specific structure not represented in the training data (Reynolds and McDonell, 2021; Min et al., 2022). Intuitively, few-shot exemplars fail to provide enough signal to learn custom outputs since those formats were designed for specialized tasks, and thus are unlikely to have appeared in open web corpora typically used to train LLMs.

Alternatively, limited data issues can also be tackled through data augmentation techniques, including altering tokens at the surface level (Wei and Zou, 2019) or mapping seed data into a latent

state before generating new examples (Sennrich et al., 2016). What these methods lack though is control over the generation process. Specifically, two aspects of control are critical when synthesizing data: *label preservation* and *diversity*. Label preservation ensures the generated utterances remain faithful to the original attributes in the seed data. Diversity ensures the generated utterances provide better coverage of the target distribution to guide the model toward better generalization.

To avoid the pitfalls of naive data augmentation, we take advantage of the LLM's ability to generate fluent text by leveraging it as a tool for controlled data generation. Concretely, we start by tuning a set of parameter-efficient soft prompts for each of the attributes present in the seed data. We then introduce a novel method for combining the **M**ixture of **S**oft **P**rompts (MSP) to generate diverse, class-conditioned training data in a carefully controlled manner. The synthetic data is finally used to train a smaller, downstream model on the task at hand (Figure 1). Using LLMs as data generators rather than black-box predictors provides interpretability and flexibility benefits since the synthesized data can be directly inspected for quality.

We apply MSP to three diverse NLU tasks to test its generalizability. Compared to directly prompt-tuning an LLM with few-shot data, using the LLM to augment the data instead is capable of outperforming a model of the *same size* by up to 27% (see Table 9). We additionally compare to a wide variety of competitive data augmentation and controlled text generation baselines where our method leads to superior downstream performance across all three benchmarks. Qualitative analysis and human evaluation further verify that the data generated by MSP ranks higher on measures of quality, specificity and correctness (see Table 2). Overall, our proposed method represents a novel, data-centric approach for using prompt-tuned LLMs to tackle domain-adaptive few-shot learning.

## 2 Task Formulation

Few-shot natural language understanding (NLU) can take many forms such as semantic parsing or named entity recognition. In such tasks, a model aims to understand a natural language input, but only has a limited number of training examples $x_i$ to do so. Formally, we are given a dataset $\mathcal{D}_s = (\{x_i, y_i\}^n)_s$ with $n$ training examples that all belong to some group of $s$ source domains. The

goal is to expand into a target domain $t$, given only $m$ examples in domain $t$: $\mathcal{D}_t = (\{x_j, y_j\}^m)$, where $m << n$. Real life applications of NLU are further complicated by the multi-aspect nature of the target, where a single label $y_i$ may be composed of multiple unique attributes $\{attr_a\}$.

### 2.1 Few-shot Direct Prediction

One straightforward way to approach this problem is to pre-train a large neural network that is capable of handling low-resource scenarios. Recently, LLMs have exhibited competitive performance in multiple few-shot tasks by using the limited data as exemplars for in-context learning (Sanh et al., 2022). However, direct prediction in this manner contains many drawbacks, such as lack of control over the prediction process. This motivates us to consider an alternative framework.

### 2.2 Data-Centered Alternative

Another way to deal with limited data is to perform data augmentation where the few-shot seed data is used to produce additional training examples $\mathcal{D}_{syn} = (\{x_k, y_k\}^p)$. Afterwards, all the original seed data is combined with the synthesized data $\mathcal{D}_s \cup \mathcal{D}_t \cup \mathcal{D}_{syn}$ to train a downstream model. To the extent the downstream model is significantly smaller than the original (~15x smaller in our case), this process can be viewed as knowledge distillation through data transfer.

Using LLMs as a data augmentation tool rather than a direct predictor confers multiple benefits: (a) Generated data can be inspected, which improves interpretability and explainability. (b) Supplementary modifications, such as data denoising or filtering, can be stacked on top of the generated data, which allows for more flexibility in improving performance. (c) Data can be used to train smaller, more computation-efficient models for faster inference. (d) Data is model agnostic, leading to transferability across model types (See Section 5.3). We take advantage of all these points in our method.

## 3 Mixture of Soft Prompts

### 3.1 Overview

Our method follows a three-step process of soft-prompt tuning, data generation, and downstream model training (see Fig 2). The full prompt fed into the LLM can be broken down into four components: instruction prefix, attribute soft prompts, domain meta-data and retrieved exemplars.

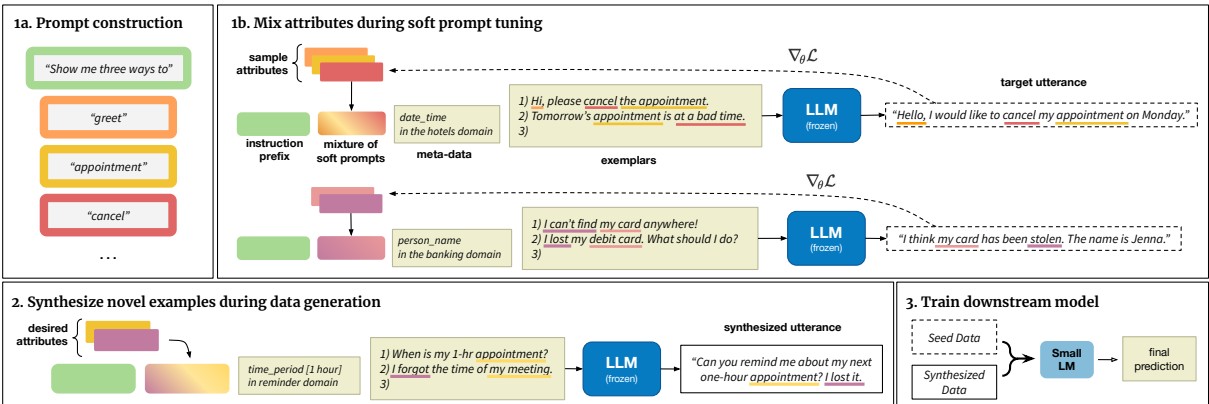

Figure 2: The instruction prefix (green) and attribute soft prompts (yellow, orange, red) are initialized (top left) then tuned (top right) using few-shot data from the target domain, while keeping the LLM unchanged. Attribute soft prompts are mixed before being fed into the LLM, with training signal expressed along dotted lines. During inference (bottom), the prompt-tuned attributes are used to generate novel examples in a controlled manner.

## 3.2 Prompt Construction

Soft prompt tuning has emerged as a parameter-efficient method for leveraging the power of LLMs without the onerous computation requirements of training from scratch (Lester et al., 2021). The core of our approach relies on soft prompts, but rather than tuning the prompts to make a direct prediction, we instead instruct the LLM to generate high quality training examples.

The full input contains four parts. (1) The first is an instruction prefix initialized with the phrase "Show me three distinct utterances that all express the X" which is shared across all training examples. (2) The soft prompts are initialized with the name and description of attribute, e.g. "song is a musical song or melody", and are often domain-specific. (3) The third part includes meta-data relevant to the task such as slot-values or domain name. (4) Finally, a handful of exemplars are appended to guide the model towards better data augmentation, selected based on attribute overlap. (See Appendix B for details.) The seed example itself is used as the target utterance for training. (Top half of Fig 2)

## 3.3 Attribute Mixing

To control the characteristics of the synthesized text, we condition the model on the desired attributes during generation. However, prior works on controlled text generation mainly focus on a single attribute constraint, such as sentiment (Qian et al., 2022a). In contrast, individual examples in our tasks all contain multiple attributes. For example, one task is multi-aspect intent detection, where a single dialogue utterance may contain three intent

attributes (Figure 2, Box 1a). How should these attribute embeddings be combined?

We experiment with five different methods of composing attributes for data generation. For all methods, we initialize a set of soft prompts $SP$ for each attribute $attr$ in the ontology of the dataset. Given a training example utterance $x_i$ and its set of attributes $y_i$, we compose a mixed prompt $\mathcal{P}_i$ that combines all relevant attributes. To do so, we draw the attribute embeddings $\{attr\_emb\} \subseteq SP$ such that $\forall attr_a \in y_i$, $attr\_emb_a$ is the attribute prompt embedding corresponding to $attr_a$. Prompt composition is then performed through one of the five following attribute mixing methods.

Suppose a seed example consists of $n$ attributes, indexed by $a$. The **Concat** method simply concatenates all attribute embeddings together and places the result in between the instruction prefix and the remaining input text.

$$\mathcal{P}_i = [attr\_emb_1; attr\_emb_a; attr\_emb_n] \quad (1)$$

A key drawback of Concat is that a variable number of attribute embeddings produces a variable-length input. To circumvent this inconvenience, we also test a **Pooling** method which combines each attribute embedding by taking the mean value across the embedding dimensions. In doing so, the fixed output dimensions allow for easy batch processing.

$$\mathcal{P}_i = \frac{1}{N} \sum_{a=1}^{N} attr\_emb_a \quad (2)$$

The limitation with simply averaging the embeddings is that it treats all embedding values equally. To see how we can combine information in a more

meaningful manner, we explore additional methods that learn to weight the attribute embeddings.

The **Attention** mechanism method begins by averaging input embeddings along the embedding dimension, passing them to a feed-forward linear layer, and going through a SiLU activation function (Elfwing et al., 2018). Layer norm followed by a temperature modulated softmax then produces an attention score $\alpha_i$. The attention score calculates a weighted sum of the attribute soft prompts, resulting in a final mixed prompt.

$$\begin{aligned}
\bar{q} &= meanpool(attr\_emb_a) \\
p &= SiLU(\mathbf{W}_q \cdot \bar{q}^T) \\
\alpha^{attn} &= softmax(LN(p)) \\
\mathcal{P}_i &= \alpha^{attn} \cdot \bar{q}
\end{aligned} \tag{3}$$

Inspired by Asai et al. (2022), who use soft prompts for knowledge sharing, we also test a modification to the Attention method that introduces a **Bottleneck** layer into the process. More specifically, the averaged input embeddings are down-projected into a smaller dimension, and followed by a non-linearity before being projected back up to the original input embedding shape. This is followed by layer norm and softmax as before to calculate the attention score.

$$\begin{aligned}
\mathbf{H}_{down} &= \mathbf{W}_{down}^T(\bar{q}) \\
\mathbf{H}_{up} &= \mathbf{W}_{up}^T(SiLU(\mathbf{H}_{down})) \\
\hat{\alpha}^{attn} &= softmax(LN(\mathbf{H}_{up})) \\
\mathcal{P}_i &= \hat{\alpha}^{attn} \cdot \bar{q}
\end{aligned} \tag{4}$$

Lastly, the **CNN** mixture method combines multiple soft prompts via a convolutional neural network. We start by padding the attribute to a fixed length. Then we pass the embedding through two layers of convolutions, where a ReLU activation function is used between each layer.

$$\begin{aligned}
q_{cnn} &= pad(attr\_emb_a) \\
q_{cnn} &= conv_1(q_{cnn}) \\
\mathcal{P}_i &= conv_2(ReLU(q_{cnn}))
\end{aligned} \tag{5}$$

### 3.4 Data Denoising

As the final step in the MSP pipeline, we take advantage of the fact that augmented data can be easily manipulated to further improve the data quality. Specifically, we over-generate 20% more data and then apply filtering to denoise the samples, bringing the number of examples back in line with the original amount. Filtering is accomplished by looping

| Dataset | Domain | Train | Dev | Test |
|---|---|---|---|---|
| NLU++ | Hotels generic | 644 | 66 | 66 |
| | Hotels specific | 300 | 34 | 34 |
| | Banking generic | 1594 | 172 | 172 |
| | Banking specific | 238 | 32 | 32 |
| CrossNER | Politics | 200 | 541 | 651 |
| | Science | 200 | 450 | 543 |
| | Music | 100 | 380 | 456 |
| | Literature | 100 | 400 | 416 |
| | AI | 100 | 350 | 431 |
| | General | 15k | 3.5k | 3.7k |
| TOPv2 | Weather | 176 | 147 | 5682 |
| | Reminder | 493 | 337 | 5767 |
| | Others | 84k | 12k | 22k |

Table 1: Number of examples included per domain in the three NLU datasets we study, divided across splits.

through the synthesized examples and dynamically choosing which ones to keep based on two factors.

The first factor is motivated by the observation that certain attribute classes are over-represented in the seed data. Thus, we sample examples at a rate which is inversely proportional to how often an attribute occurs. In effect, this re-balances the data so all attributes have an equal chance of appearing.

The second factor aims to improve label preservation by lowering diversity. During inspection of preliminary results, we found that most errors came as a result of low correctness because the generated data deviated too far away from the target label. Thus, we counteract this by keeping synthesized examples that are more similar to the original seed data. Similarity between synthesized and original data is measured using cosine similarity of their SentenceTransformer (Reimers and Gurevych, 2019) embeddings. We found this method of keeping similar synthesized examples to work better than using a lower temperature during generation.

## 4 Experimental Setup

### 4.1 Datasets and Tasks

We test on three diverse, multi-attribute natural language understanding datasets. These datasets offer pre-determined few-shot splits and natural division of source and target domains for testing.

**NLU++** Our first task is multi-aspect intent detection (Casanueva et al., 2022), where a model should predict all intents present in the given dialogue turn. Since it is possible to predict too many or too few intents, success is measured by F1 score. Two topics, hotels and banking, are in the dataset,

with both containing generic and specific versions. This effectively yields four blocks of examples. The cross-domain setting in the paper evaluates on the generic version of each topic, so we set three blocks as the source domains (ie. hotel generic, hotel specific, banking specific) and leave the fourth block as the target domain (ie. banking generic). The target output is in the form of a list of intents.

**CrossNER** Our second task is cross-domain named entity recognition, where the main challenge is transferring from the general news domain to one of five specialized domains with custom entity types (Liu et al., 2021b). For example, Politics contains unique *politician* and *election* entities not found in other domains. The target output is a series of (entity category, entity value) pairs.

**TOPv2** The third task we examine is compositional, task-oriented semantic parsing (Chen et al., 2020). The generic source data consists of six individual domains: Alarm, Event, Messaging, Navigation, Music, and Timer. The two target domains are Reminder and Weather, whose training datasets allow only 25 SPIS (samples per intent and slot). Following the setup of the original paper, we also perform preprocessing steps to build the canonical form for prediction. The final format consists of multi-intent labels followed by slot-value pairs.

## 4.2 Baseline Methods

**Direct Prediction** We use FLAN-T5 XXL (Chung et al., 2022) as our base model for generating data. GODEL (Peng et al., 2022) serves as our smaller downstream LLM, which starts with a T5 backbone (Raffel et al., 2020) and is fine-tuned on dialogue related data. At just 770M parameters, the student model contains roughly 15 times fewer parameters than the teacher. For a fair comparison, we use the exact same model again to make direct predictions with a prompt-tuning-only setup.

We also compare against billion-parameter models optimized through in-context learning and chain-of-thought prompting (Wei et al., 2022), namely GPT-3.5-turbo and GPT-4 [1]. The in-context learning prompt consists of four components: an instruction prefix; a comprehensive list of domain-specific attributes; relevant meta-data; and five question-answer exemplars. The exemplars are selected based on the frequency of their attributes

across the dataset, with the top-ranked candidates chosen as representative examples. The instruction prefix was manually prompt engineering across dozens of attempts to ensure fairness. We perform chain-of-thought prompting by breaking apart the task into 3 steps. First, we ask the model to predict the domain of the sentence. Next, we have the model think about what attribute types are present in the utterance. Finally, we have the LLM directly predict the attribute values of the given input.

**Data Augmentation** To improve upon the vanilla student model, we augment the few-shot data with various techniques. We first try the very simple Easy Data Augmentation (EDA) (Wei and Zou, 2019) which randomly drops and swaps tokens. We also consider masked in-filling (Kobayashi, 2018) that masks out certain portions of text and uses a BERT model to fill them back in. We also look into BART-large model (Lewis et al., 2020) trained on paraphrase corpora (Dolan and Brockett, 2005; Zhang et al., 2019). Finally, we also compare against round-trip translation (RTT) across five pivot languages (Sennrich et al., 2016). These techniques all generate diverse data, but may not be accurate since they have no mechanism to enforce attribute labels to appear in the synthesized data.

**Controlled Text Generation** We also test methods that condition on the attributes during generation to encourage label preservation. We consider a Conditional LM (CLM), which fine-tunes GPT2 to produce an example utterance when given a serialized representation of the attribute label (Anaby-Tavor et al., 2020). Another direction performs weighted decoding of the logits during inference, where we use DExperts for constrained text generation (Liu et al., 2021a). We also consider a conditional variational auto-encoder (CVAE) (Hu et al., 2017; Xia et al., 2020) that learns to generate attribute constrained outputs by sampling from the latent space between the encoder and decoder. We additionally examine a lexically-motivated baseline, Keyword2Text (K2T) where the probability distribution is shifted toward target keywords during decoding time (Pascual et al., 2021). Lastly, we experiment with a prompt-tuning baseline (PT) that uses GPT-4 to generate synthetic examples. This includes an option that applies our denoising technique (PT + Denoising) before training on the downstream task. All rhese techniques exhibit a stricter adherence to labels, but may lack diversity.

---

[1] Due to the high cost of using GPT-4, we run experiments for just one domain per dataset.

## 4.3 Automatic Evaluation

Beyond downstream accuracy, we also evaluate the synthesized data quantitatively with three metrics. *Distinct@K* measures the diversity of text based on unique n-grams, where we set k=1,2,3 following common practice (Li et al., 2016). *Perplexity* represents text fluency, which we measure through GPT2-large (Radford et al., 2019). Third, we use *Correctness* to check how well the synthetic data preserves the proper attribute labels. To do so, we train an oracle with all available data (ie. no longer few-shot) to classify the primary attributes within an utterance. (More details in Appendix A)

## 4.4 Implementation Details

The instruction prefix is set to a length of 100 tokens, while the attribute token length is set to 20 tokens. After hyper-param tuning, we settle on 3e-2 as the learning rate for the teacher model and 3e-5 for the student [2] (See Appendix B). Augmentation methods are standardized to generate four new datapoints per seed example.

## 5 Results and Analysis

### 5.1 Main Results

As seen in Table 3, MSP w/ Bottleneck achieves state-of-the-art results across all three datasets, with an average 20.3% improvement over the original baselines. MSP reaches the highest end-to-end scores on 8 out of 9 possible domains, while also demonstrating competitive performance on the one remaining domain (ie. TOPv2 Reminder). Notably, the one area where MSP does not achieve the highest rank, it is actually surpassed by a meta-learning technique. However, meta-learning and data augmentation are orthogonal methods, so in practice, these two can and should be combined to produce even better results than either alone.

Despite the remarkable ability of large pre-trained LMs to generate coherent and fluent language, their capacity to produce structured outputs is limited. In particular, performance from direct prediction models deteriorate when dealing with utterances featuring more complex structures and diverse attributes, such as TOPv2. Leveraging a billion-parameter model brings marginal improvement, but performance is still well below data synthesis baselines. As seen in Table 4, even when the LLM is scaled to the size of GPT-4 (OpenAI,

---

[2]Our code and further implementation details can be found at https://github.com/derekchen14/mixture_soft_prompts.

---

|  | Quality | Specificity | Accuracy |
|---|---|---|---|
| Paraphrase | 3.4 | 3.1 | 4.4 |
| CLM | 3.6 | 3.3 | 4.2 |
| MSP | 4.4 | 4.3 | 4.7 |

Table 2: Human evaluation results comparing three methods on a 5-point Likert scale. MSP ranks highest across all metrics with average Fleiss $\kappa = 0.72$.

2023), direct prediction yields worse performance than MSP. On the other hand, by leveraging LLMs for data augmentation, our method successfully leans into the innate strengths of language models as text generators rather than forcing them to learn specialized target output sequences.

Compared to the data augmentation baselines, MSP consistently leads to better results across all three datasets. Qualitatively, we observe that all four naive augmentation methods produce examples which ultimately deviate away from the desired semantic attributes. This causes a degradation in downstream performance compared to MSP (See Table 11).

While the controlled text generation (CTG) methods are able to outperform the naive GODEL baseline, CTG underperforms the same GODEL model by an average of 10% when augmented with MSP synthesized data. This performance trend is reflected even when using GPT-4 for controlled text generation, as shown in Table 4. Qualitative analysis reveals the CTG methods are unable to pick up the complex structure required by the multi-attribute generation task, so this synthesized data ends up as noise and actually hurts performance. On the other hand, MSP is able to reliably handle lexical, semantic and structural constraints.

### 5.2 Synthesized Data Quality

For human evaluation, we surveyed 30 fluent English speakers who reviewed the generated utterances according to given metrics: (1) Quality - the text is grammatically correct, coherent and fluent. (2) Specificity - the text is specific to the target topic, rather than generically applicable. (3) Accuracy - the text correctly reflects the desired semantic attributes. The results can be seen in Table 2. Testing against the top DA method (Paraphrase) and the top CTG method (CLM), our method (MSP) ranks highest across all metrics, with particularly large improvements in Specificity.

Beyond downstream accuracy, we also use automatic evaluation to judge the quality of the synthe-

| | Method | Hotels | Banking | Politics | Science | Music | Literature | AI | Reminder | Weather |
|---|---|---|---|---|---|---|---|---|---|---|
| Original Baselines | LSTM-based † | 67.3 | 49.2 | 61.5 | 52.1 | 51.7 | 48.4 | 45.2 | 45.8 | 65.1 |
| | (Ro)BERTa-based † | 79.3 | 74.2 | 68.7 | 69.4 | 68.3 | 63.6 | 58.9 | 63.7 | 76.0 |
| | Transformer-based † | 75.4 | 65.2 | 70.4 | 66.8 | 72.1 | 67.1 | 60.3 | 70.5* | 77.7* |
| Direct Prediction | Few-Shot (GODEL) | 74.5 | 64.2 | 76.7 | 72.0 | 75.8 | 69.2 | 64.7 | 60.5 | 77.1 |
| | ICL (GPT-3.5-turbo) | 52.4 | 52.1 | 54.4 | 57.2 | 67.2 | 52.9 | 48.4 | 39.0 | 69.6 |
| | ICL (FLAN-T5) | 59.0 | 60.5 | 27.6 | 26.8 | 13.8 | 34.7 | 56.9 | 55.1 | 63.2 |
| | Prompt-tune (GPT-J-6B) | 68.2 | 59.5 | 63.9 | 52.4 | 62.9 | 62.9 | 52.3 | 49.5 | 69.7 |
| | Prompt-tune (FLAN-T5) | 81.2 | 69.2 | 71.6 | 68.0 | 69.3 | 58.3 | 57.1 | 50.6 | 72.9 |
| Data Augmentation | Easy Data Augmentation | 81.8 | 68.0 | 77.3 | 72.1 | 76.3 | 71.0 | 65.4 | 62.5 | 80.6 |
| | Round Trip Translation | 77.3 | 71.2 | 75.3 | 71.0 | 74.1 | 66.3 | 63.0 | 54.8 | 77.6 |
| | Paraphrasing (BART) | 76.6 | 64.9 | 79.9 | 72.6 | 75.8 | 71.0 | 65.7 | 61.0 | 81.7 |
| | Masked In-Fill | 78.2 | 69.6 | 79.7 | 74.2 | 76.8 | 70.3 | 65.3 | 63.8 | 78.5 |
| Controlled Text Generation | Conditional LM | 76.8 | 71.5 | 75.7 | 73.9 | 76.2 | 69.8 | 63.9 | 67.8 | 80.2 |
| | DExperts | 83.1 | 71.8 | 78.2 | 71.4 | 72.6 | 66.3 | 63.6 | 50.9 | 69.7 |
| | Conditional VAE | 77.0 | 70.4 | 74.0 | 69.6 | 72.9 | 67.9 | 64.3 | 46.0 | 69.1 |
| | Keyword2Text | 74.8 | 67.8 | 74.8 | 72.4 | 75.3 | 68.2 | 62.3 | 64.0 | 81.0 |
| Our Method | MSP w/ Concat | 83.5 | 84.2 | 77.3 | 72.8 | 71.2 | 66.3 | 62.7 | 62.2 | 82.9 |
| | MSP w/ Pooling | 83.8 | 82.9 | 80.6 | 73.0 | 79.0 | 71.9 | 66.7 | 62.8 | 83.9 |
| | MSP w/ Attention | 86.5 | 84.1 | 78.2 | 73.9 | 77.1 | 72.1 | 65.7 | 64.5 | 83.3 |
| | MSP w/ Bottleneck | 89.7 | 84.6 | 79.5 | 74.8 | 80.0 | 69.6 | 65.9 | 65.1 | 84.6 |
| | MSP w/ Convolution | 85.0 | 80.3 | 80.5 | 73.6 | 78.7 | 67.8 | 65.0 | 61.0 | 83.2 |

Table 3: End-to-end F1-scores for NLU++, CrossNER, and TOPv2. †Original baseline scores are aggregated from previous works (See Section 4.1). *BART Copy-Ptr with meta-learning. For exact model types, please refer to the appendix. Different MSP mixtures achieve state-of-the-art in all domains except one.

| | | Hotels | Music | Weather |
|---|---|---|---|---|
| Direct Prediction | ICL | 67.7 | 68.4 | 72.1 |
| | CoT | 75.4 | 66.5 | 67.9 |
| Controlled Text Gen | PT only | 76.9 | 72.3 | 75.8 |
| | PT + denoise | 77.1 | 70.3 | 78.6 |

Table 4: GPT-4 underperforms MSP for NLU++ Hotels, CrossNER Music, and TOPv2 Weather domains. The first two rows show direct prediction using in-context learning and prompt-tuning on GPT-4. The latter two rows show end-to-end scores using GPT-4 to generate additional training data. PT is short for Prompt-tuning.

sized data. DA methods have lower linguistic quality, but achieve higher attribute conservation since changing a few tokens can easily harm readability, but usually do not change the overall meaning. In contrast, CTG methods generally exhibit high fluency, but their low correctness scores also reveal a difficulty in capturing all the desired semantic attributes. Table 6 shows that MSP generated data strikes a balance between diversity and correctness, without the need for any manual tuning.

### 5.3 Ablations

In order to understand the impact of each part in the MSP pipeline, we run ablations along one domain from each dataset. Results in Table 7 show that all parts of our technique provide meaningful gains. In particular, the instruction prefix tells the model to generate similar examples, and removing this

prompt consistently leads to the lowest scores. As expected, removing any part of the trainable soft prompts leads to substantial degradation.

The last row in Table 7 includes one final tweak to the method which swaps out the Flan-T5 backbone for GPT-J-6B (Wang and Komatsuzaki, 2021). This change from a sequence-to-sequence model to a causal language model highlights the flexibility of our approach since the transferrability of data is agnostic to its provenance. Although the downstream model trained with GPT-augmented data is not as strong as the model trained with T5-augmented data, it does clearly outperform the GPT-J-6B model performing direct prediction.

## 6 Related Work

Our paper tackles multi-aspect few-shot learning where the target is a structured output containing multiple attributes, and the model only has a few examples from which to learn such a pattern. Given the complexity of the task, previous research trained custom models for each dataset (Zheng et al., 2022; Schucher et al., 2022). Instead, we leverage the power of LLMs (Brown et al., 2020; Chung et al., 2022) to design a more general solution through controlled data synthesis.

In particular, we compose multiple *Soft Prompts* to generate the desired training data for the downstream task. Consequently, we build upon foundational work studying soft prompt tuning (Lester

| Dataset | Attributes and Meta-data | Generated Text |
|---|---|---|
| NLU++ | Intents: *'change', 'booking'* 
 Domain: Hotels | **Original**: change booking 
 **Generated**: I have a reservation that I need to modify |
| CrossNER | Entity categories: *'artist'*; 
 Entities: *'John O 'Reilly', 'Bobby Rondinelli', 'John Miceli', 'Chuck Burgi'* 
 Domain: Music | **Original**: Chuck Burgi ( 1991-1992 , 1992-1995 , 1996-1997 ) , John Miceli ( 1992 , 1995 ) , John O 'Reilly ( 1995-1996 ) and Bobby Rondinelli ( 1997-2004 ) . 
 **Generated**: He was followed as musical artist by Chuck Burgi (1992-1995, 1996-1997 ), John Miceli ( 1992, 1995 ), John O 'Reilly ( 1995-1996 ) and Bobby Rondinelli ( 1997-2004 ). |
| TOPv2 | Intents: *'help_reminder'* 
 Domain: Reminder | **Original**: How does the reminder notification sound when it plays out loud? 
 **Generated**: Set a reminder for my dentist appt |

Table 5: Qualitative examples of synthesized data applying our method (MSP) on different datasets. The generated utterances typically demonstrate the preservation of the desired semantic attributes and lexical entities of the target domain, as seen in NLU++ and CrossNER. We select an example from TOPv2 to highlight where MSP struggles.

| | NLU++ | | | CrossNER | | | TOPv2 | | |
|---|---|---|---|---|---|---|---|---|---|
| | Distinct@1/2/3 | Correct. | Perpl. | Distinct@1/2/3 | Correct. | Perpl. | Distinct@1/2/3 | Correct. | Perpl. |
| EDA | 8.80 / 45.1 / 69.7 | 94.8 | 1.33 | 19.8 / 53.6 / 68.9 | 81.3 | 1.07 | 12.3 / 47.7 / 70.7 | 73.0 | 1.45 |
| RTT | 7.90 / 29.4 / 48.9 | 94.7 | 1.24 | 15.3 / 33.5 / 42.6 | 81.2 | 1.05 | 11.4 / 33.8 / 53.3 | 75.5 | 1.34 |
| Paraphrase | 6.40 / 22.4 / 36.8 | 92.2 | 1.23 | 12.8 / 26.1 / 31.4 | 89.9 | 1.06 | 9.55 / 26.1 / 40.6 | 76.0 | 1.31 |
| In-Fill | 5.45 / 23.5 / 38.8 | 93.8 | 1.36 | 10.2 / 23.6 / 29.1 | 91.9 | 1.06 | 9.15 / 27.1 / 41.1 | 69.8 | 1.54 |
| CLM | 2.00 / 5.90 / 10.5 | 76.8 | 1.27 | 11.9 / 23.7 / 29.7 | 86.4 | 1.07 | 7.05 / 18.9 / 29.7 | 79.4 | 1.43 |
| DExperts | 4.60 / 13.1 / 21.9 | 53.4 | 1.15 | 7.74 / 21.6 / 30.1 | 22.9 | 1.03 | 2.35 / 4.85 / 7.10 | 62.3 | 1.27 |
| CVAE | 0.25 / 0.80 / 2.20 | 19.2 | 2.00 | 53.9 / 73.0 / 81.6 | 68.2 | 1.09 | 5.10 / 16.2 / 28.1 | 71.5 | 1.45 |
| MSP | 3.90 / 13.95 / 23.4 | 98.0 | 1.27 | 14.5 / 32.7 / 40.7 | 93.6 | 1.06 | 7.10 / 20.9 / 33.9 | 74.8 | 1.43 |

Table 6: Automatic evaluation results with Distinct@K measuring diversity, Correctness measuring label preservation and Perplexity representing text fluency. Correct and Perpl are short for correctness and perplexity, respectively.

| | Hotels | Literature | Reminder |
|---|---|---|---|
| MSP w/ Bottleneck | 89.7 | 68.0 | 65.1 |
| - data denoising | 88.5 | 67.1 | 64.9 |
| - instruction | 82.2 | 63.4 | 62.4 |
| - meta-data | 89.4 | 67.3 | 64.2 |
| - attribute prompt | 87.6 | 64.1 | 63.8 |
| - exemplars | 88.8 | 65.5 | 62.2 |
| + GPT-J-6B | 84.5 | 63.9 | 63.7 |

Table 7: Ablation studies about the impact of individual components in the pipeline of MSP on the downstream task performance. The first row is the baseline with all components under bottleneck mixing method. The (-) sign indicates the absence of a specific step.

et al., 2021; Vu et al., 2022), as well as other parameter efficient fine-tuning methods (Houlsby et al., 2019; Li and Liang, 2021). Alternatively, Wang et al. (2022) and Chen et al. (2022) perform data augmentation with prompting, but their prompts are not compositional since their task setups are focused on single-aspect class prediction.

*Data Augmentation* is a common technique in NLP for counteracting the limited data available with few-shot learning (Feng et al., 2021; Chen and Yin, 2022). Flavors of data augmentation include surface form alteration (Wei and Zou, 2019),

latent perturbation (Sennrich et al., 2016; Fabius et al., 2015) or auxiliary supervision (Chen and Yu, 2021). Our method can be considered a form of text generation with transformers (Kumar et al., 2020; Ng et al., 2020), which lately rely on increasingly larger language models (Yoo et al., 2021; Wang et al., 2021a,b). Whereas these methods paraphrase or pseudo-label a seed utterance, MSP instead conditions on a label to control the generation of text.

As a result, our method is also related to *Controlled Text Generation* techniques. Unlike constrained lexical decoding (Pascual et al., 2021), which aims to produce text that contains a pre-specified set of keywords, our work is focused on controlling the semantics of the output, such as a topic or user intent (Mou et al., 2016; Hokamp and Liu, 2017; Post and Vilar, 2018; Yao et al., 2019). For semantic control, a wide range of options exist for guiding generation towards a single attribute, including those that train a model from scratch (Keskar et al., 2019; Wang et al., 2019) or those which only tune a few parameters (Ribeiro et al., 2021; Lin et al., 2021; Yu et al., 2021; Liu et al., 2023). There are even methods that keep the base model frozen and instead manipulate the

logits with weighted decoding to control the output (Dathathri et al., 2020; Krause et al., 2021; Yang and Klein, 2021; Zhang and Song, 2022). These methods can stay on topic, but often sacrifice the ability to generate specific tokens, while MSP is able to maintain both semantic and lexical control, yielding superior results.

Lastly, MSP is related to techniques that combine multiple prompts together. Nayak et al. (2022) propose combining soft prompts through concatenation, but their aim is to improve direct prediction in a vision-based task, as opposed to data generation for NLU tasks. Qin and Eisner (2021) target classification tasks using pre-trained LMs, but their mixture-of-experts technique selects individual prompts from multiple candidates to satisfy a single constraint, rather than mixing multiple prompts to meet multiple constraints. Our work is most similar to those that perform multi-aspect text generation (Yang et al., 2022; Gu et al., 2022; Qian et al., 2022b). However, the primary aim of improving text quality aligns with our secondary aim (Subsection 5.2). Whereas these prior efforts focus exclusively on text generation, our method controls generation as a means to an end.

# 7  Conclusion

Our paper presents an alternative method for few-shot learning using LLMs as an intermediate data augmentation tool rather than for direct prediction. By performing data generation as an intermediate step for improved end-to-end task performance, our method yields such benefits as interpretability, flexibility and modularity. Compared against other strong data augmentation methods, we show that MSP yields higher quality data that can be effectively used to improve performance in downstream tasks. This parameter-efficient method to perform controlled data generation is a powerful paradigm for using LLMs in low-resource scenarios; the positive results from the methods proposed in this work suggest promising future work in exploring tighter controls and smarter filtering mechanisms for data augmentation. Ultimately, we encourage others to consider use LLMs as a tools for generating data, rather than only for generating direct predictions.

## 8 Limitations

The largest model we could feasibly access is a T5-XXL which contains 11 billion parameters. While we did test against GPT-3.5 and GPT-4, it is entirely feasible that a GPT5 (unknown size) or OPT3 (175 billion), or PALM model (540 billion) may outperform our results in these few-shot settings using prompt-tuning with exemplars. However, we would posit that as the ability of these truly gigantic models improve, their ability to generate superior training data would also improve in concert, so our method would still be worthwhile. Evidence for this hypothesis is seen from the transition from GPT-J-6B to T5-XXL, which leads to better prompt-tuning results along with better MSP results. However, we cannot know for sure at the moment without access to more compute resources.

The other major limitation of our work is the lack of a clear optimization target during data generation. We used BLEU score of the synthesized example compared to the original seed example as a proxy for measuring model convergence. However, it turns out that achieving a higher BLEU score during MSP training does not always translate to superior downstream results. Ideally, we would be able to directly leverage the downstream accuracy as a training signal back to optimizing the MSP model, which we leave to as future work.

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

# A Oracle Correctness Classifier

In order to perform automatic evaluation on correctness, we train an oracle attribute classifier based on DeBERTa-XLarge (He et al., 2021). However, there is a chicken-and-egg problem since the goal of the downstream task is also to predict attributes. To get around this issue, we make three key simplifying assumptions. To begin, we use all available data for training, rather than limiting to the few-shot setting. For example, we go from 25 SPIs to over 1000 SPIs on the TOPv2 dataset. Since this classifier is meant to operate an an oracle rather than a display of model capability, we even include examples from the development set for training. Secondly, we only focus on the main attribute (intent) rather than second-order details (slots). Finally, we simplify the attribute prediction task by classifying each attribute individually, rather than in a compositional manner. Whereas our final model must perform the task by sequentially generating the label, we take advantage of the ontology to turn this into a classification task over a finite number of labels. In doing so, we obtain a correctness classifier that is able to reach over 90% accuracy across all target domains (See Table 13).

# B Training Setup Details

To select the exemplars, we first represent each example in the seed data by its attributes. Then, for each of those examples, we sort the available training data by amount of overlap with the representation to find the top 10 closest in attribute alignment. From those top-ranked candidates, we randomly sample two examples to serve as exemplars during data generation. We tested with 1, 2, or 3 exemplars and found that k=2 typically worked best, although 1 exemplar was not far behind.

We use a parameter of 4, for the num generations. When training the MSP model, we tested the learning rates from [1.0, 0.3, 0.1, 0.03]. We found that the range of rates were much higher than expected. For final tests, we used lr=0.3. We train with an effective batch size of 24, for example with batch-size flag set to 8 and gradient accumulation set to 3 steps. However, if the GPU runs out of memory, we might lower this to 6 and 4, respectively. For fine-tuning the downstream task on GODEL large model, we test the learning rate across [1e-4, 3e-5, 1e-5, 3e-6] for up to 14 epochs with early stopping, and found 3e-5 worked best.

# C Generalization to Out-of-Domain

The motivation behind this work is to leverage controllable data synthesis as a means of expanding products and services into target domains where labeled data is scarce. By definition, these new areas are out-of-domain (OOD) for a model trained only on source domains. Our strategy for generalizing to OOD spaces is to perform data augmentation.

Successful data augmentation ideally helps a model generalize on a test set given a limited training set by expanding the seed data to cover the entire solution space. As a result, reliably controlling this process is akin to automating data collection. Following the principles of active learning, ideal data collection involves selecting examples to label that provide high coverage and high impact (Settles, 2009). Turning back to data augmentation, these goals translate to promoting diversity and label preservation, as mentioned in Section 1.

Our method (MSP) has a number of levers to pull in order to increase diversity. One idea is to simply increase the temperature parameter of the LLM during text generation. Another path is to shuffle the exemplars used for guidance or change the way the exemplars are retrieved. Building upon this, one could even exclude the seed example from the exemplars to minimize the copying behavior common to LLMs. A particularly exciting direction to pursue is composing novel attribute combinations not seen in the seed set. For example, one utterance might have 'greet' and 'change' intents in an airline domain (e.g. *Hello, I would like to change my flight*), while a second utterance contains 'request_info' and 'when' intents for e-commerce (e.g. *Can you tell me when the store opens?*). These could be remixed to generate a wholly new utterance with 'request_info' and 'change' intents in the restaurant domain (e.g. *I'd like to know how I can change my reservation*). We actually tested all these ideas during preliminary experimentation. As it turns out, they don't help.

In fact, we found that label preservation quickly overtook diversity in terms of being the most important factor for influencing downstream impact. Consequently, we actually had to take painstaking efforts to dial *down* the diversity. We lowered temperature parameters. We selected very similar exemplars for guidance. We only used attribute combinations that were found in the seed data. And we even added a denoising step to minimize variation (Subsection 3.4). Although limiting diversity

worked in our case, we believe this is largely due to the static nature of our test sets, while the distribution of real life test environments exhibit an ever-expanding long tail. In either case, the flexibility of MSP allows the practitioner to choose what they want, whether that's precision within a specific area or robustness to cover OOD.

## D   Mixing Other Parameter-Efficient Tuning Methods

Our core idea for mixing adapter weights is flexible enough to accommodate other parameter-efficient tuning methods such as LoRA (Hu et al., 2022). Our key contribution is that the mixing should be learned rather than relying on prompt engineering. To this point, we ran additional experiments on NLU++ hotel, CrossNER music and TOPv2 weather by mixing LoRA weights where each adapter matrix represents a single attribute. Specifically, we use the bottleneck method and replace the attention projection linear layer with a corresponding LoRA linear layer. The results were $86.4$, $76.7$ and $80.1$, respectively, which outperform other non-mixture baselines but do underperform MSP. We did not have much time to tune the results, but note that this experiment shows how learning to mix already outperforms most other baselines.

| | Method | Politics | Science | Music | Literature | AI |
|---|---|---|---|---|---|---|
| Baselines | BiLSTM-CRF | 56.6 | 50.0 | 44.8 | 43.0 | 43.6 |
| | Coach LSTM | 61.5 | 52.1 | 51.7 | 48.4 | 45.2 |
| | BERT-tagger | 68.7 | 69.4 | 68.3 | 63.6 | 58.9 |
| | Multi-Cell-LSTM | 70.6 | 66.4 | 70.5 | 67.0 | 58.3 |
| | LST-NER | 70.4 | 66.8 | 72.1 | 67.1 | 60.3 |
| Domain Adaptive Pre-training | Multi-Cell-LSTM + DAPT | 71.5 | 67.7 | 74.2 | 68.6 | 61.6 |
| | LST-NER + DAPT | 73.2 | 70.1 | 76.8 | 70.8 | 63.3 |
| | Span-Integration + DAPT | 72.1 | 68.8 | 75.7 | 69.0 | 62.6 |
| Direct Prediction | Few-Shot (GODEL) | 76.7 | 72.0 | 75.8 | 69.2 | 64.7 |
| | In-context Learn (GPT-3.5) | 54.4 | 57.2 | 67.2 | 52.9 | 48.4 |
| | In-context Learn (FLAN-T5) | 27.6 | 26.8 | 13.8 | 34.7 | 56.9 |
| | Prompt-tune (GPT-J-6B) | 63.9 | 52.4 | 62.9 | 62.9 | 52.3 |
| | Prompt-tune (FLAN-T5) | 71.6 | 68.0 | 69.3 | 58.3 | 57.1 |
| Data Augmentation | Easy Data Augmentation | 77.3 | 72.1 | 76.3 | 71.0 | 65.4 |
| | Round Trip Translation | 75.3 | 71.0 | 74.1 | 66.3 | 63.0 |
| | Paraphrasing (BART) | 79.9 | 72.6 | 75.8 | 71.0 | 65.7 |
| | Masked In-Fill | 79.7 | 74.2 | 76.8 | 70.3 | 65.3 |
| Controlled Text Generation | Conditional LM | 75.7 | 73.9 | 76.2 | 69.8 | 63.9 |
| | DExperts | 78.2 | 71.4 | 72.6 | 66.3 | 63.6 |
| | Conditional VAE | 74.0 | 69.6 | 72.9 | 67.9 | 64.3 |
| | Keyword2Text | 74.8 | 72.4 | 75.3 | 68.2 | 62.3 |
| Our Method | MSP w/ Concat | 77.3 | 72.8 | 71.2 | 66.3 | 62.7 |
| | MSP w/ Pooling | **80.6** | 73.0 | 79.0 | 71.9 | **66.7** |
| | MSP w/ Attention | 78.2 | 73.9 | 77.1 | **72.1** | 65.7 |
| | MSP w/ Bottleneck | 79.5 | **74.8** | **80.0** | 69.6 | 65.9 |
| | MSP w/ Convolution | 80.5 | 73.6 | 78.7 | 67.8 | 65.0 |

Table 8: End-to-end F1-scores for CrossNER. Different MSP mixtures achieve state-of-the-art in all domains.

| | Hotels | Banking |
|---|---|---|
| ConveRT | 75.4 | 65.2 |
| LM12-1B | 67.3 | 49.2 |
| RoB-base-QA | 79.3 | 74.2 |
| AlBERT-base-QA | 76.7 | 72.7 |
| Few-shot (GODEL) | 74.5 | 64.2 |
| In-context learn (GPT-3.5) | 52.4 | 52.1 |
| In-context learn (FLAN-T5) | 59.0 | 60.5 |
| Prompt tune (GPT-J-6B) | 68.2 | 59.5 |
| Prompt tune (FLAN-T5) | 81.2 | 69.2 |
| Easy Data Augmentation | 81.8 | 68.0 |
| Round Trip Translation | 77.3 | 71.2 |
| Paraphrasing (BART) | 76.6 | 64.9 |
| Masked In-Fill | 78.2 | 69.6 |
| Conditional LM | 76.8 | 71.5 |
| DExperts | 83.1 | 71.8 |
| Conditional VAE | 77.0 | 70.4 |
| Keyword2Text | 74.8 | 67.8 |
| MSP w/ Concat | 83.5 | 84.2 |
| MSP w/ Pooling | 83.8 | 82.9 |
| MSP w/ Attention | 86.5 | 84.1 |
| MSP w/ Bottleneck | **89.7** | **84.6** |
| MSP w/ Convolution | 85.0 | 80.3 |

Table 9: Full end-to-end F1-scores evaluated on generic splits of hotel and banking domains of NLU++.

| | Reminder | Weather |
|---|---|---|
| Transfer learn (LSTM) | 45.8 | 65.1 |
| Transfer learn (RoBERTa) | 63.7 | 76.0 |
| Few-shot (T5-Large) | 50.2 | 68.2 |
| Fine tune (BART-CopyPtr) | 55.7 | 71.6 |
| Joint training (BART-CopyPtr) | 58.9 | 74.7 |
| Transfer learn (BART-CopyPtr) | 68.0 | 75.9 |
| Meta-Learn (BART-CopyPtr) | **70.5** | 77.7 |
| Few-shot (GODEL) | 60.5 | 77.1 |
| In-context learn (GPT-3.5) | 39.0 | 69.6 |
| In-context learn (FLAN-T5) | 55.1 | 63.2 |
| Prompt tune (GPT-J-6B) | 49.5 | 69.7 |
| Prompt tune (FLAN-T5) | 50.6 | 72.9 |
| Easy Data Augmentation | 62.5 | 80.6 |
| Round Trip Translation | 54.8 | 77.6 |
| Paraphrasing (BART) | 61.0 | 81.7 |
| Masked In-Fill | 63.8 | 78.5 |
| Conditional LM | 67.8 | 80.2 |
| DExperts | 50.9 | 69.7 |
| Conditional VAE | 46.0 | 69.1 |
| Keyword2Text | 64.0 | 81.0 |
| MSP w/ Concat | 62.2 | 82.9 |
| MSP w/ Pooling | 62.8 | 83.9 |
| MSP w/ Attention | 64.5 | 83.3 |
| MSP w/ Bottleneck | 65.1 | **84.6** |
| MSP w/ Convolution | 61.0 | 83.2 |

Table 10: Full end-to-end accuracy evaluated on reminder and weather target domains within TOPv2.

| | | |
|---|---|---|
| **Easy Data Augmentation (EDA)** | | |
| NLU++ | Intents: *'restaurant'*, *'when'*
Domain: Hotels | **Original**: when do you start serving dinner?
**Generated**: do when dinner start serving you? |
| TOPv2 | Intents: *'delete_reminder'*;
Slots types: *'person_reminded'*, *'todo'*
Domain: Reminder | **Original**: Delete my reminder to pay ATAP business taxes.
**Generated**: Delete my reminder to ATAP business taxes. |
| **Round Trip Translation (RTT)** | | |
| CrossNER | Entity categories: *'miscellaneous'*, *'algorithm'*, *'task'*;
Entities: *'PCA'*, *'non-negative matrix factorization'*, *'K-NN'*, *'Feature extraction'*, *'NMF'*, *'linear discriminant analysis'*, *'LDA'*, *'canonical correlation analysis'*, *'CCA'*, *'dimension reduction'*, *'principal component analysis'*, *'feature vectors'*, *'pre-processing step'*
Domain: AI | **Original**: Feature extraction and dimension reduction can be combined in one step using principal component analysis ( PCA ) , linear discriminant analysis ( LDA ) , canonical correlation analysis ( CCA ) , or non-negative matrix factorization ( NMF ) techniques as a pre-processing step followed by clustering by K-NN on feature vectors in reduced-dimension space.
**Generated**: As a step, extraction and size reduction techniques can be combined using the analysis of the main components (PCA), linear discrimination analysis (LDA), canonic correlation analysis (CAS) or non-negative matrix factorization (NMP) as a preliminary processing phase, followed by the grouping of K-NN on vectors in a space with a reduced dispersibility factor. |
| TOPv2 | Intents: *'create_reminder'*, *'get_todo'*, *'get_recurring_date_time'*;
Slots types: *'person_reminded'*, *'frequency'*, *'todo'*, *'recurring_date_time'*, *'date_time'*
Domain: Reminder | **Original**: Remind me daily to buy a present for Papa before July 28th.
**Generated**: Remind me to buy my father a present every day before July 28th. |
| **Paraphrase** | | |
| CrossNER | Entity categories: *'party'*, *'organization'*, *'politician'*
Entities: *'Red-Green coalition'*, *'Bundestag'*, *'Renate Künast'*, *'Joschka Fischer'*, *'Jürgen Trittin'*, *'SPD'*
Domain: politics | **Original**: Despite losses for the SPD , the Red-Green coalition government commanded a very slight majority in the Bundestag and was renewed , with Joschka Fischer as foreign minister , Renate Künast as minister for consumer protection , nutrition and agriculture , and Jürgen Trittin as minister for the environment .
**Generated**: Despite losses for the SPD, the Red-Green coalition government commanded a very slight majority in the Bundestag and was re-elected, with Joschka Fischer as foreign minister, Renate Künast as minister for consumer protection, nutrition and agriculture and Jürgen Trittin as Minister for the environment. |
| TOPv2 | Intents: *'unsupported_weather'*;
Slots types: *'location'*
Domain: Weather | **Original**: Any tornado warnings in my area?
**Generated**: What is the tornado warning in my area? |
| **In-Fill** | | |
| NLU++ | Intents: *'thank'*, *'cancel_close_leave_freeze'*
Domain: Hotels | **Original**: thanks, I'll leave the room before 7:25 a.m..
**Generated**: thanks, I'll leave the office before 7:25 a.m.. |
| CrossNER | Entity categories: *'product'*;
Entities: *'Octave'*, *'MATLAB'*
Domain: AI | **Original**: Octave helps in solving linear and nonlinear problems numerically , and for performing other numerical experiments using a that is mostly compatible with MATLAB.
**Generated**: Octave helps in solving linear and nonlinear problems numerically , and for performing other numerical experiments using a that is fully compatible with MATLAB . |

Table 11: The synthetic samples generated by Data Augmentation (DA) methods. The orange words highlight wrong attributes, whereas the green words represent correctly generated attributes. Paraphrase performs well since it is able to maintain attribute meanings. Despite the clear decrease in fluency, EDA also maintains attribute labels and greatly assists downstream performance. RTT drops key entities in the pivot language, leading to noisy data.

**Controlled Language Model (CLM)**

| | | |
|---|---|---|
| TOPv2 | Intents: *'update_reminder'*, *'get_recurring_date_time'* 
 Slot types: *'todo'*, *'recurring_date_time_new'*, *'frequency'* 
 Domain: Reminder | **Original**: Change the reminder to go the store for psa to every Wednesday 
 **Generated**: Move the reminder to go the store for psa to every Wednesday. |
| CrossNER | Entity categories: *'event'*, *'award'*; 
 Entities: *'Grand Prix Spécial du Jury'*, *'Palme d 'Or"*, *'Cannes Film Festival'*, *'FIPRESCI prize'* 
 Domain: Literature | **Original**: The film was presented at the Cannes Film Festival , won the Grand Prix Spécial du Jury and the FIPRESCI prize , and was nominated for the Palme d 'Or. 
 **Generated**: He won the Cannes Film Festival's Grand Prix Spécial du Jury and the FIPRESCI prize, as well as the Palme d 'Or. |

**Decoding-Time Expert (DExperts)**

| | | |
|---|---|---|
| NLU++ | Intents: *'cancel_close_leave_freeze'*, *'direct_debit'* 
 Domain: banking | **Original**: cancel my direct debit to LA VN Kitchen 
 **Generated**: I need to cancel my booking at 10:00 a.m. as I have a private meeting with my wife. |
| TOPv2 | Intents: *'get_recurring_date_time'*, *'update_reminder'*; Slots types: *'todo'*, *'recurring_date_time_new'*, *'frequency'* 
 Domain: Reminder | **Original**: Update the reminder about medication to once a day. 
 **Generated**: Change the reminder to pay the bill on the 15th of every month to the 15th of every month. |

**Conditional Variational Auto-Encoder (CVAE)**

| | | |
|---|---|---|
| CrossNER | Entity categories: *'book'*, *'writer'*; 
 Entities: *'Gulliver 's Travels"*, *'An Argument Against Abolishing Christianity'*, *'Swift'*, *'A Modest Proposal'*, *'A Tale of a Tub'* 
 Domain: Literature | **Original**: Swift is remembered for works such as A Tale of a Tub ( 1704 ) , An Argument Against Abolishing Christianity ( 1712 ) , Gulliver 's Travels ( 1726 ) , and A Modest Proposal ( 1729 ). 
 **Generated**: Swift is remembered for a Tubation's works in the 17th century, An Argument Abolberg, Gulliver at A Modest Abishing and Robert Gposal, music. |
| CrossNER | Entity categories: *'election'*, *'organization'*, *'country'*, *'miscellaneous'*; 
 Entities: *'Independence and Democracy'*, *'MEPs'*, *'2004 European Parliament election in Denmark'*, *'EDD'*, *'2004 European Parliament election in Poland'*, *'UK'*, *'Sweden'*, *'2004 European Parliament election in the United Kingdom'* 
 Domain: Politics | **Original**: Following the 2004 European Parliament election in the United Kingdom , 37 MEPs from the UK , 2004 European Parliament election in Poland , 2004 European Parliament election in Denmark and Sweden founded a new European Parliamentary group called Independence and Democracy as a direct successor to the EDD group. 
 **Generated**: Following the 2004 European Parliament election in the United Kingdom, 37 MEPs from Denmark from the UK and Sweden European Parliament election in the United Kingdom general election in Poland. |

Table 12: Qualitative examples for Controlled Text Generation (CTG) methods. The orange words highlight wrong attributes, whereas the green words represent correctly generated attributes. Despite largely staying on topic and grabbing a few keywords, CVAE and DExperts tend to lose lexical control, leading to poor performance.

| | NLU++ | | CrossNER | | | | | TOPv2 | |
|---|---|---|---|---|---|---|---|---|---|
| | Hotels | Banking | Politics | Science | Music | Literature | AI | Weather | Reminder |
| Learning rate | 1e-4 | 1e-4 | 1e-5 | 1e-5 | 1e-5 | 1e-5 | 1e-5 | 1e-5 | 1e-5 |
| Batch Size | 10 | 10 | 8 | 10 | 10 | 10 | 10 | 8 | 8 |
| Accuracy | 97.0 | 94.5 | 93.0 | 94.8 | 95.5 | 94.1 | 92.4 | 99.3 | 96.5 |

Table 13: Results from training the correctness classifier