# OpenReview forum: "Mixture of Soft Prompts for Controllable Data Generation"
_EMNLP/2023/Conference — EMNLP 2023 Findings_

### Official Review · Reviewer_rBGC · 2023-08-02

**Soundness:** 4

**Excitement:**

4: Strong: This paper deepens the understanding of some phenomenon or lowers the barriers to an existing research direction.

**Paper Topic And Main Contributions:**

This paper focuses on the problem of predicting multi-attribute outputs with the help of generative language models in a domain-transfer setting. Rather than make the models generate predictions directly, the authors propose an approach to use them for generating data that is then used to train a task- and domain-specific smaller model. The data generation process relies on a Mixture of Soft Prompts (MSP) that is trained using data in the target domain, before being then leveraged to generate new data - also in the target domain. Through several experiments the authors show that using an MSP data augmentation approach helps improve over several types of baselines, and achieves state-of-the-art results on several of the datasets that they use in their evaluation.

**Reasons To Accept:**

Overall, this is a well written paper with an interesting and relevant premise - namely using LLMs as data augmenters rather than predictors for classes of problems that require complex structured outputs. As the authors point out, this comes with a few advantages such as interpretability, flexibility, efficiency and transferability. The proposed MSP approach to controlling generations is an interesting, and - to the best of my knowledge - novel one. And the experiments are generally well structured, demonstrating that the proposed approach does improve on the baselines they compare against.

**Reasons To Reject:**

Unfortunately, there are some real concerns (which to be fair the authors do touch on in their discussion of 'Limitations'). To begin with, the paper talks about using LLMs, but actually uses (and compares against, with the exception of GPT-3.5Turbo) models that by today's standards are not particularly *large*. While the authors argue that larger models - that they haven't been able to test due to access issues - may also be better data generators and therefore maintains the relevance of their contributions, they fail to account for the fact that these models may be good enough predictors that data generation is no longer needed for these multi-attribute outputs.

But even if we put that concern aside, and grant that data augmentation does yields smaller models with certain benefits, there is no clear reason to believe that truly large LMs require a strategy such as MSP to yield multi-attribute structured outputs. Without evidence of these failings on the part of LLMs that the authors haven't tested, the contributions in the paper - while interesting - could be short-lived or even irrelevant already.

More specific to the experiments they have conducted, the authors do not compare against the capabilities of LLMs as predictors with prompt engineering strategies generally accepted to yield better results, such as simple chain-of-thought prompting or (since this consists of multi-attribute predictions) prompt-chaining.

**Reproducibility:**

4: Could mostly reproduce the results, but there may be some variation because of sample variance or minor variations in their interpretation of the protocol or method.

**Reviewer Confidence:**

4: Quite sure. I tried to check the important points carefully. It's unlikely, though conceivable, that I missed something that should affect my ratings.

---

> ### Author Rebuttal · Authors · 2023-08-28
>
> We thank the reviewer for recognizing the novelty of our approach, as well as the benefits of interpretability, flexibility, efficiency and transferability.  To give a bit of a background, when we first published our paper, GPT-4 was not yet released.  With that said, we have recently gained access to the API with results listed in our general comment.  Despite improved results over GPT-3.5, the larger GPT-4 model still does not outperform our MSP technique. This highlights the shortcomings of current LLMs in composing and mixing precise information, providing fertile ground for future research.

---

### Official Review · Reviewer_uw25 · 2023-08-04

**Soundness:** 4

**Excitement:**

3: Ambivalent: It has merits (e.g., it reports state-of-the-art results, the idea is nice), but there are key weaknesses (e.g., it describes incremental work), and it can significantly benefit from another round of revision. However, I won't object to accepting it if my co-reviewers champion it.

**Paper Topic And Main Contributions:**

This paper proposes a controlled text generation method (MSP) by tuning soft prompts. Compared to previous works, MSP allows for integrating various attributes and leverages the generation capacity of large models to achieve more controllable data generation. The effectiveness of the approach is demonstrated through experiments.

**Reasons To Accept:**

The paper explores several methods for composing attributes in soft prompts and takes a denoising step to improve the data quality. The experiments are comprehensively conducted by selecting a diverse set of baselines and performing detailed comparisons with different methods. The effectiveness of the approach is demonstrated on three datasets.

Update：Overall, the work of the article is reasonable. I increased my Soundness score to 4.

**Reasons To Reject:**

Models like GPT-3.5 can directly produce data containing various attributes with appropriate instructions. Therefore, it seems more cumbersome to optimize soft prompts in training for data generation. The paper lacks a comparison with this direct prompting way.

**Reproducibility:**

4: Could mostly reproduce the results, but there may be some variation because of sample variance or minor variations in their interpretation of the protocol or method.

**Reviewer Confidence:**

4: Quite sure. I tried to check the important points carefully. It's unlikely, though conceivable, that I missed something that should affect my ratings.

---

> ### Author Rebuttal · Authors · 2023-08-28
>
> We thank the reviewer for noting the effectiveness of the approach on all three datasets.  The main concern is around using GPT-3.5 for generating augmented data.  In our added comment above, we run exactly the experiment of generating data for one domain in each of our datasets, then use this data to train our downstream GODEL model.  As seen in the results, MSP still outperforms GPT data augmentation because prompt engineering is difficult and imprecise.

---

### Official Review · Reviewer_4Rzr · 2023-08-04

**Soundness:** 4

**Excitement:**

3: Ambivalent: It has merits (e.g., it reports state-of-the-art results, the idea is nice), but there are key weaknesses (e.g., it describes incremental work), and it can significantly benefit from another round of revision. However, I won't object to accepting it if my co-reviewers champion it.

**Paper Topic And Main Contributions:**

This paper focuses on utilizing the LLM as a data augmentation tool to enhance the performance of small-scale models on multi-attribute natural language understanding (NLU) tasks. Instead of prompting LLM or adapting LLM with few-shot approaches, the proposed MSP pipeline leverages parameter-efficient transfer learning (PETL) modules to generate synthesized training data for downstream models using the LLM.
The MSP pipeline has two key highlights:
1) MSP enables efficient control over the characteristics of the synthesized data. Specifically, the PETL modules (soft prompts) are initially trained with attribute-specific data and then used to generate multi-attribute synthesized data instances with a certain mixture-of-experts (MoE) approach.
2) Quality control of augmentation data. Two identified factors, based on empirical observations, are applied to filter the over-generated data in order to improve data quality.

The authors validate the effectiveness of MSP by fine-tuning the downstream GODEL model with synthesized data generated by the FLAN-T5 teacher. The results demonstrate that MSP outperforms the baselines and other data augmentation methods across three multi-attribute NLU tasks. Furthermore, both automatic and human evaluations indicate that the MSP pipeline excels in terms of synthesized data quality.

**Questions For The Authors:**

A. Have you applied data filtering to other data augmentation baselines to ensure comparison equity?

B. Regarding the second factor of data filtering, could you please provide more details on how you evaluated the similarity between the synthesized and original data? Is there a specific threshold set to the correctness metric?

C. Are there any measures taken to ensure the quality of human evaluation? Did the evaluators receive reasonable allowance?

**Reasons To Accept:**

- **Comprehensive Empirical Study**: This work includes an extensive range of experiments to validate the effectiveness of each proposed method. The study encompasses different NLU tasks, such as multi-aspect intent detection, cross-domain named entity detection, and task-oriented semantic parsing. Furthermore, the research compares multiple methods from Direct Prediction, Data Augmentation, and Controlled Text Generation as baselines, ensuring that each part of the overall pipeline outperforms related methods.

- **Efficiency**: For the green NLP topic, this work highlights the significance of utilizing powerful LLMs for data augmentation while considering computational costs. Recognizing the practical constraints of the real world, the proposed MSP pipeline focuses on distilling LLM's knowledge for a small LM in a parameter-efficient manner. This approach allows for a more realistic and sustainable usage of the LLM.

**Reasons To Reject:**

- **Detailed Discussion**: The paper lacks detailed explanations for certain methodological choices, which may raise concerns about the rigor of the study. Specifically, 1) in Section 3.2, the authors use a particular instruction prefix, but the reason behind this choice remains unclear. 2) While the use of Prompt Tuning (Lester et al., 2021) is mentioned as a strategy for building soft prompts, the advantages of this approach over other PETL methods such as LoRA and adapter tuning are not discussed. 3) In Section 4.3, the rationale for selecting GPT2-large to evaluate text fluency instead of other PLMs is not adequately justified.

- **Novelty**: While the paper presents a strong empirical study, its theoretical novelty may be limited. The components of the MSP pipeline, such as PETL, MoE, and prompt engineering, are established methods in their respective research directions. Although the MSP pipeline achieves promising performance, it appears to be a composition of different existing methods rather than introducing substantially new innovations. As such, the uniqueness of the MSP pipeline may be limited.

**Update:** I increased my soundness score during the rebuttal phase. Please refer to the discussion below.

**Reproducibility:**

4: Could mostly reproduce the results, but there may be some variation because of sample variance or minor variations in their interpretation of the protocol or method.

**Reviewer Confidence:**

4: Quite sure. I tried to check the important points carefully. It's unlikely, though conceivable, that I missed something that should affect my ratings.

**Typos Grammar Style And Presentation Improvements:**

- Figure 1: the horizontal line (LLM-> controlled generated data) is skewed.
- Table 4,5,6: Please use the bold to highlight the important results.
- Line 230, 244: Layer norm -> Layer normalization
- Line 295, 297, 384: ie. -> i.e.
- Line 378, 401: k -> $K$

---

> ### Author Rebuttal · Authors · 2023-08-28
>
> We would like to start by thanking the reviewer for recognizing the key merits of the paper including the realistic application of LLMs and the comprehensiveness of the experiments. We now dive a bit into MSP's novelty.
>
> MSP is composed of two parts: (1) *controllable* data augmentation and (2) specialized data denoising. Given an arbitrary NLP task, MSP can produce high quality training data for that task. To the extent that the downstream task continues to exhibit low accuracy after training, we can use MSP again to target specific attributes of the task that are not solved. MSP is the only feasible method that exists to run such a  continually improving ML system because our method is both time-efficient and highly reliable.  In contrast, training a model through a LM-objective requires millions of dollars in compute and months to train.  Prompt engineering is certainly efficient too, but using an LLM for direct prediction or data generation is unreliable since prompts are too brittle.  As detailed in the general comment, this drawback exists even when the LLM is GPT-4. This highlights how control and consistency are still within the purview of fine-tuning alone. We hope our work encourages further study on (efficient methods of) supervised learning, distinct from progress towards improved prompt engineering.
>
> To address the questions:
>  - _Applying data denoising to other baselines_: The data denoising techniques are part of the novelty of our method since they are unique to MSP.  In either case, we apply the exact same data denoising to the GPT-4 augmented data, where this actually leads to a drop in performance. (See Part D in Official Comment)
>  - _More details on data filtering_: The similarity between the synthesized and original data is cosine similarity of their embeddings, as encoded by a SentenceTransformer. No threshold hyper-param is necessary to tune since we pre-determine the augmentation factor of 4x (line 392) for fair comparison. Suppose we start with 100 seed examples and aim to generate 400 augmented examples. Subsequently, we rank all synthesized examples and simply keep the 400 closest ones based on cosine similarity.
>  - _Quality of human evaluation_: To ensure quality of evaluation, we (1) wrote detailed explanations of the three human eval metrics (2) tested with a small pilot audience to iterate on explanations to make sure they were clear and (3) sent out the survey to the larger group. The participants were researchers from our affiliated labs acting as volunteers.
>
> To address the detailed explanation for methodological choices:
>  - _In Section 3.2, why choose this particular instruction prefix?_ Initializing the instruction-prefix with natural language text just provides a starting point.  Since the soft-prompt is ultimately learned, any reasonable instruction would suffice.  In fact, we tried 2-3 variations in preliminary testing and found them all to work reasonably well. We are happy to add more results to the appendix for this if desired.
>  - _Why use Prompt Tuning over LoRA or other forms of adapter tuning?_ To be honest, LoRA wasn’t as widely discussed back in late 2022 when we started working on this.  Our core idea for mixing adapter weights is flexible enough though to accommodate LoRA as well.  Our key contribution is that the mixing should be learned rather than relying on prompt engineering. To this point, we ran additional experiments on NLU++ hotel, CrossNER music and TOPv2 weather by mixing LoRA weights where each adapter matrix represents a single attribute. Specifically, we use the bottleneck method and replace the [attention projection linear layer](https://github.com/huggingface/transformers/blob/main/src/transformers/models/llama/modeling_llama.py#L256) with a corresponding [lora linear layer](https://github.com/huggingface/peft/blob/6c44096c7b8d55a2ecf24be9bc68393467e1584a/src/peft/tuners/lora.py#L834). The results were 86.4 ( 89.7 for soft prompts), 76.7 (80.0 for soft prompts) and 80.1 (84.6 for soft prompts).  We didn’t have much time to tune the results, but notably learning to mix already outperforms most other baselines.
>  - _In Section 4.3, why select GPT2-large to evaluate text fluency?_ Giant models take lots of compute resources to run, so our rationale was to choose the smallest model possible that could still offer meaningful perplexity scores. We felt GPT2-large was a reasonable choice following other related works (e.g. [DisCup](https://arxiv.org/abs/2210.09551), Zhang and Song)
>
> We will certainly include all this information and address the typos as suggested.  Thank you for the thorough review!

---

### Meta-Review · Area_Chair_wrwC · 2023-09-18

**Recommendation:** 4

**Metareview:**

The paper introduces a novel approach, Mixture of Soft Prompts (MSP), for controllable data generation using Large Language Models (LLMs) in the context of multi-attribute natural language understanding tasks. Instead of relying on LLMs for direct prediction, MSP leverages LLMs as data augmentation tools, enhancing the training of smaller, domain-specific models. The paper demonstrates the effectiveness of MSP through comprehensive experiments and shows it outperforms baselines on multi-attribute NLU tasks.

The reviewers raised concerns about (1) the model size used in the experiments and whether truly large LLMs might not require MSP; (2)
the absence of evidence showcasing limitations in large LLMs for multi-attribute predictions; (3) the lack of a comparison with prompt engineering strategies like simple chain-of-thought prompting or prompt-chaining.

The authors have partially addressed the first concern by gaining access to GPT-4 and reporting that even GPT-4 does not outperform MSP. However, they have not fully addressed the second and third concerns, as they have not provided evidence or comparisons with prompt engineering strategies. Thus, while the paper presents a novel approach, some concerns raised by the reviewers remain unaddressed.

---

### Decision · Program_Chairs · 2023-10-07

**Decision:**

Accept-Findings

**Comment:**

The paper introduces a novel approach, Mixture of Soft Prompts (MSP), for controllable data generation using Large Language Models (LLMs) in the context of multi-attribute natural language understanding tasks. Instead of relying on LLMs for direct prediction, MSP leverages LLMs as data augmentation tools, enhancing the training of smaller, domain-specific models. The paper demonstrates the effectiveness of MSP through comprehensive experiments and shows it outperforms baselines on multi-attribute NLU tasks.

The reviewers raised concerns about (1) the model size used in the experiments and whether truly large LLMs might not require MSP; (2)
the absence of evidence showcasing limitations in large LLMs for multi-attribute predictions; (3) the lack of a comparison with prompt engineering strategies like simple chain-of-thought prompting or prompt-chaining.

The authors have partially addressed the first concern by gaining access to GPT-4 and reporting that even GPT-4 does not outperform MSP. However, they have not fully addressed the second and third concerns, as they have not provided evidence or comparisons with prompt engineering strategies. Thus, while the paper presents a novel approach, some concerns raised by the reviewers remain unaddressed.